# Comparison of the Capacitance of a Cyclically Fatigued Stretch Sensor to a Non-Fatigued Stretch Sensor When Performing Static and Dynamic Foot-Ankle Motions

**DOI:** 10.3390/s22218168

**Published:** 2022-10-25

**Authors:** Andrea Karen Persons, Carver Middleton, Erin Parker, Will Carroll, Alana Turner, Purva Talegaonkar, Samaneh Davarzani, David Saucier, Harish Chander, John E. Ball, Steven H. Elder, Chartrisa LaShan Simpson, David Macias, Reuben F. Burch V.

**Affiliations:** 1The Ohio State Wexner Medical Center, Jameson Crane Sports Medicine Institute, Columbus, OH 43202, USA; 2Department of Human Factors & Athlete Engineering, Human Performance Lab, Center for Advanced Vehicular Systems, Mississippi State University, Starkville, MS 39759, USA; 3Department of Electrical & Computer Engineering, Mississippi State University, Starkville, MS 39762, USA; 4Department of Kinesiology, Mississippi State University, Starkville, MS 39762, USA; 5Department of Industrial & Systems Engineering, Mississippi State University, Starkville, MS 39762, USA; 6Department of Agricultural and Biological Engineering, Mississippi State University, Starkville, MS 39762, USA; 7OrthoVirginia, 1920 Ballenger Ave., Alexandria, VA 22314, USA

**Keywords:** stretch sensor, high-cycle fatigue, capacitance, cyclic softening, drift, ankle joint, smart sock

## Abstract

Motion capture is the current gold standard for assessing movement of the human body, but laboratory settings do not always mimic the natural terrains and movements encountered by humans. To overcome such limitations, a smart sock that is equipped with stretch sensors is being developed to record movement data outside of the laboratory. For the smart sock stretch sensors to provide valuable feedback, the sensors should have durability of both materials and signal. To test the durability of the stretch sensors, the sensors were exposed to high-cycle fatigue testing with simultaneous capture of the capacitance. Following randomization, either the fatigued sensor or an unfatigued sensor was placed in the plantarflexion position on the smart sock, and participants were asked to complete the following static movements: dorsiflexion, inversion, eversion, and plantarflexion. Participants were then asked to complete gait trials. The sensor was then exchanged for either an unfatigued or fatigued plantarflexion sensor, depending upon which sensor the trials began with, and each trial was repeated by the participant using the opposite sensor. Results of the tests show that for both the static and dynamic movements, the capacitive output of the fatigued sensor was consistently higher than that of the unfatigued sensor suggesting that an upwards drift of the capacitance was occurring in the fatigued sensors. More research is needed to determine whether stretch sensors should be pre-stretched prior to data collection, and to also determine whether the drift stabilizes once the cyclic softening of the materials comprising the sensor has stabilized.

## 1. Introduction

Motion capture (MOCAP) is currently the gold standard for assessing movement of the human body [1]; however, MOCAP is not always readily available to clinicians and, as a laboratory environment, may not always accurately represent natural situations, especially those that lead to injury. Due to the limitations of MOCAP, methods to collect movement data, such as joint angles, in a non-clinical setting are desirable. To overcome the limitations of measuring joint angles in a non-clinical setting, video or photographic-based goniometer smartphone applications have been developed and validated to measure joint angles [2,3], personalized medical approaches using additive manufacturing (3D printing) techniques to print sensors using existing or novel materials that can be tailored to suit the needs of a particular patient are being explored [4,5,6,7], and research is ongoing in the development of a smart sock that utilizes capacitive stretch sensors to measure ankle joint angles in real-time during athletic events [8,9,10,11,12,13,14].

Within the context of sports performance, sensors can be used to capture real-time data regarding: (1) training, (2) competition, (3) safety, and (4) recovery [15]. Stretch sensors are particularly suited for biomechanical and orthopaedic measurements as they are typically placed along the joints to produce the requisite stretch needed to generate a signal [8,15,16]. Because of their ability to capture joint angles, stretch sensors may also be beneficial in applications beyond sports performance–applications such as assessing the progress of a patient recovering from orthopaedic surgery, assessing the progress of a patient rehabilitating from an injury, in patients who have conditions that affect their mobility due to injury, disease, or age-related changes, or for monitoring at-home adherence to rehabilitation protocols [8,9,10,16,17,18,19,20,21,22].

Measurements taken during athletic events will not only allow researchers to determine the average joint angles that occur during various sports, but, in the case of injury, will also capture the sequence of joint angles preceding the injury. Such data may be used by physical therapists and athletic trainers to improve the biomechanics of athletes and prevent injury [15], or by orthopaedic surgeons to improve the tensioning of tendons and ligaments during surgical interventions. For example, ankle sprains are a common injury with a global incidence of approximately 27,000,000 cases per year [23]. Although all ankle sprains do not result from athletic activities, use of a smart sock utilizing stretch sensors to measure ankle joint angles during rigorous athletic activities could help clinicians and researchers understand the types of movements that lead to ankle sprains, and this information could be further used for the development of footwear that aids in the prevention of ankle sprains for not only athletes, but also for general consumers.

To produce reliable measurements, two considerations are key. First is the placement of the stretch sensors about the joint, and bony landmarks can be used as a guide to ensure consistent placement of the sensors [8]; therefore, determining the correct placement of the sensors also requires an understanding of the anatomy and biomechanics of the selected joint. By correctly placing the sensors, replicable signal patterns are produced, and deviations in the signal pattern can be recognized. Such deviations in the signal pattern may help to identify underlying joint pathologies. Secondly, the sensors must be durable. For example, to move the sock prototype from the laboratory to the basketball court, the sensors must be able to withstand low-cycle fatigue (LCF). Basketball players are estimated to take 1260 running steps per game [24]; however, 32 basketball games may be played in one regular season by a National Collegiate Athletics Association (NCAA) Division I (D1) basketball team exclusive of the conference and national tournaments, subjecting a sensor to approximately 40,320 cycles in a regular season moving the sensors into the high-cycle fatigue (HCF) regime [14]. Most fatigue studies of stretch sensors only capture LCF, and the durability of the materials and signal beyond a few thousand cycles remains uncertain [14]. Further, the biomechanics of the chosen joint influence the forces placed upon the stretch sensors, and when designing quasi-static and fatigue testing methods to examine the material properties and durability of the sensors, these forces should be taken into consideration. As the current study is a continuation of research into the development of a smart sock to measure joint angles of the human ankle, the loading conditions used during the materials testing are based on loading conditions of the human ankle joint.

### 1.1. Overview of the Human Foot-Ankle Complex

The human ankle joint is comprised of three major joints which include the subtalar, tibiotalar, and talocalcaneonavicular joints, while the foot itself contains 26 individual bones that articulate along 33 joints (Figure 1) [25,26]. Allowing for inversion and eversion of the foot, the subtalar joint consists of the talus and calcaneus bones which are primarily linked by the interosseous talocalcaneal ligament [25]. Essentially a mortise and tenson, the tibiotalar joint connects the tibia and fibula of the lower leg to the talus bone of the foot and acts as a hinge joint allowing for plantarflexion and dorsiflexion [25]. Working in concert, the medial malleolus of the tibia, and the lateral malleolus of the fibula help to stabilize the tibiotalar joint, while the anterior inferior tibiofibular ligament (AITL) provides the primary structural support for the ankle [25,26]. Similar to the subtalar joint, the talocalcaneonavicular joint also allows for inversion and eversion of the foot. The talocalcaneonavicular joint is comprised of the talus, navicular, and calcaneus bones [25].

#### Range of Motion (ROM) of the Human Ankle Joint

Platarflexion is the movement of the foot away from the leg, and can reach a maximum 50°–55° angle [25,26]. Conversely, dorsiflexion is the movement of the foot towards the leg and typically has a maximum angle of 20°. Inversion is the medial movement of the foot and typically peaks at an angle between 5° and 10°, while eversion is the lateral movement of the foot. Maximum eversion produces an approximately 5° angle (Figure 2) [26].

### 1.2. Quasi-Static and Fatigue Testing of StretchSense™ StretchFABRIC Sensors

The Athlete Engineering team at Mississippi State University is currently conducting research into the ability of the commercially available StretchSense™ StretchFABRIC sensors to capture joint angles and ROM of the foot-ankle complex during athletic practices and competitions [8,10,11,12,16]. A key requirement for any stretch sensor is the ability to produce reliable data both for their intended purpose and for the timeframe of usage [15]; however, the material properties and the electromechanical fatigue properties of the StretchFABRIC sensors remain unclear. A stretch sensor applied to measure joint angles and ROM during athletic practices and competitions would be subjected to HCF over the course of a season [14], but would not be cycled continuously as during a fatigue test. Between practices and competitions, the stretch sensors have an opportunity for the materials to relax and the signal to recover; however, the reliability of the signal produced after incurring HCF conditions remains unknown.

To investigate the electromechanical properties of the StretchSENSE™ StretchFABRIC sensors, the sensors were first subjected to quasi-static tensile testing to determine if the material strain and capacitive output were linearly correlated. Tensile testing was chosen as the preferred force to investigate because plantarflexion both provides the greatest degree of motion for the ankle [8,26], and results in extension (stretching) of the sensor. The sensors were then subjected to high cycle fatigue (HCF) testing. Capacitance data was simultaneously captured throughout both the quasi-static and fatigue tests with the expectation that the peak and valley strain as measured by the fatigue instrumentation and the peak and valley capacitance as measured by the sensor would be strongly correlated and overlap when plotted. The signal from the fatigued StretchFABRIC sensors was then compared to an unfatigued StretchFABRIC by interchanging the sensors on the smart sock prototype and having participants perform static and dynamic ankle movements while wearing the sock.

## 2. Materials and Methods

This study utilizes StretchSense™ StretchFABRIC sensors (StretchSense, Auckland, New Zealand) which are capacitive sensors comprised of a proprietary fluidic capacitive element housed within a silicone casing that has been affixed to a stretch fabric substrate via an adhesive. These sensors exploit the Poisson’s ratio of their comprising materials; whereby, as the material is stretched, the cross-sectional area decreases which is reflected as a change in capacitance [14,27]. The change in capacitance should be linearly correlated with the applied strain [27]. StretchSense sensors including the fabric, measure 128 mm L × 35 mm W × 0.56 mm thickness (Figure 3). (Production of these sensors has ceased since the onset of this project).

### 2.1. Quasi-Static Tests

All quasi-static tests were performed at ambient temperature using an Instron 5869 electromechanical testing system equipped with a 100 N load cell. Simultaneous with the quasi-static tensile tests, the capacitance (pF) of the sensors was recorded. Capacitance data were collected at 25 Hz.

Each sensor underwent five pre-test stretch/relaxation cycles at a strain rate of 1.0 mm/s to help ensure that the signal produced by the stretch/relaxation of the sensor was being captured synchronously with the mechanical test. The tensile tests began immediately following the pre-test cycles and involved stretching the sensors 60% of their gauge length. Three StretchFABRIC sensors were tested at one of three speeds: 0.6 mm/s, 1.2 mm/s, or 2.4 mm/s to determine if the material and capacitance exhibit strain-rate dependence.

Stress–strain curves for each sample were generated from the quasi-static tensile data. The stress–strain curves were then compared to one another to determine the stretch sensors exhibited strain-rate dependency. To assess reliability of the signal, the correlation of the strain and capacitance were analyzed statistically. Because the capacitance sampling rate was higher than that of the Instron, the capacitance data were decimated by a resample factor of 2.0 without the use of a filter. Normality of the strain and capacitance data were then tested using the Shapiro–Wilk test set at an α = 0.05 with the null hypothesis stating that the data are normally distributed. Based on the results of the Shapiro–Wilk test, the strain and capacitance data are not normally distributed (*p* < 0.05 for all speeds, rejection of null, data are not normally distributed) (Table 1); therefore, to determine if the strain and capacitance are correlated, a two-tailed, non-parametric Spearman’s rank correlation coefficient test was performed for the strain and capacitance data collected at each speed. The limits of the Spearman’s rank correlation coefficient test are −1 to +1, with a coefficient of −1 indicating that one variable is negatively correlated to that of the other, while a coefficient of +1 is indicative of the variables being closely correlated. Coefficients near zero indicate that no correlation exists [28]. All signal processing and statistical analyses were performed in OriginPro Version 2022 (OriginLab Corporation, Northampton, MA, USA).

### 2.2. Fatigue Tests

Two StretchSENSE™ StretchFABRIC sensors were subjected to 25,000 cycles of tensile fatigue at a frequency of 2.0 Hz using an MTS 858 Servohydraulic Table Top Tester equipped with a 25 kN load cell. A frequency of 2.0 Hz was chosen to approximate the frequency produced during human walking [29]. The capacitive output of the sensors was simultaneously recorded at a rate of 25.0 Hz throughout the fatigue testing.

To match orthopaedic loading conditions for the foot-ankle complex, data from Saucier et al. [8] who correlated plantarflexion of the ankle joint with capacitance values captured from StretchSense™ StretchFABRIC sensors, were used (Figure 4). When the capacitance was normalized to account for variation in the base capacitances of different StretchFABRIC sensors, Saucier et al. [8] found that 450 pF was correlated with 10° of plantarflexion, while 650 pF was correlated with 50° of plantarflexion; therefore, prior to the fatigue test, the base, or resting, capacitance of the unstretched sensor was determined. The first fatigued sensor registered a base capacitance of approximately 425 pF, while the base capacitance of the second sensor was approximately 410 pF.

Following the determination of the base capacitance, the sensor was manually stretched in the grips of the MTS until the capacitance reached 445 pF, and the displacement at this point was recorded and served as the return displacement for the fatigue test. The first sensor was then further stretched until the capacitance reached 650 pF, and the displacement at this point was also recorded and served as the maximum displacement that the sensor would reach; therefore, the sensor was stretched to a capacitance of 650 pF and relaxed to a capacitance of 445 pF rather than relaxation to base capacitance. As the base capacitance of the second sensor was slightly lower than that of the first sensor, to prevent excessive stretch, the second sensor was stretched until a capacitance of 600 pF was reached.

Because the capacitance dataset for each fatigued StretchFABRIC sensor contained more than 5000 capacitance values, use of the Shapiro–Wilk test for normality was contraindicated; therefore, the normality of the peak and valley strain and capacitance were tested using the Lilliefors modification of the Kolmogorov–Smirnov test at an α = 0.05 [28]. Both the strain and capacitance datasets for both sensors had non-normal distributions (*p* < 0.05, reject the null hypothesis, data are not normally distributed) (Table 2). The correlation between the peak and valley strain and the peak and valley capacitance was therefore tested using a two-tailed, non-parametric Spearman’s Rank Correlation Coefficient test.

### 2.3. Participant Trials

Twelve participants were recruited to complete static and dynamic ankle movements while wearing the smart sock prototype per Mississippi State University IRB-17-725 (Figure 5). The smart sock prototype consists of a compression sock equipped with metal hook and eye fasteners to facilitate the attachment and removal of the StretechSENSE™ StretchFABRIC sensors. Four sensors are attached to the current sock prototype. The first sensor overlies the midline of the ankle joint and measures plantarflexion, while a second sensor overlies the Achilles tendon and measures dorsiflexion. Inversion and eversion are measured by a sensor overlying the lateral malleolus and a sensor overlying the medial malleolus, respectively. The sensors are connected to a data puck via thin cables. Signals are sent from the puck to a LINUX equipped laptop by Bluetooth connection. Additionally, a cable management system of 3D printed spools is also attached to the sock to prevent excess cable from interfering with the movement of the participants. Additionally, all trials used only the smart sock prototype for the right foot.

Participants were randomly assigned to begin their trials with either the fatigued or non-fatigued plantarflexion sensor. The participants were helped to don the sock, and seated. Participants were then instructed to complete four static movements including plantarflexion, dorsiflexion, inversion, and eversion. Prior to each movement, the foot of the participant was placed in the neutral position using a manual goniometer. Each movement was completed three times for a total of 12 static movements per participant per sensor.

Following the completion of the static movements, participants were asked to walk, beginning with their right foot, 4.6 m (15 ft) between two lines marked on the floor. This distance was chosen to allow for the completion of at least three complete gait cycles. Each participant completed six gait trials resulting in 18 gait cycles per participant per sensor.

After the first set of static and dynamic trials were completed, the plantarflexion sensor was then swapped (i.e., if the participant began the trials using the fatigued plantarflexion sensor, then the second set of trials was performed using the unfatigued sensor), and both the static and dynamic trials were repeated using the swapped sensor. Following completion of the trials, the sensor data were normalized to remove any effects related to variance in the base capacitance of the sensors used in the participant trials. Data from the female participants was then averaged for each movement. The data from the one male participant was not included to remove any effect of sex on the data [30].

## 3. Results

### 3.1. Quasi-Static Testing

The stress–strain curves produced for each strain-rate are “stair-stepped,” with elastic moduli of 6.0 × 10^−7^ GPa (Figure 6A). Further, continuous stretching of the sensor to ≥80% of its gauge length results in debonding of the material housing the sensor leads from the fabric substrate.

Based on the results of Spearman’s rank correlation coefficient tests, the strain and capacitance at each speed are strongly correlated with each speed returning a coefficient of 0.99 (Table 3). Figure 6B–D show that the strain and capacitance plot close to one another and overlap in some areas of the curve.

### 3.2. Fatigue Testing

Because of the size of the load cell (25 kN) used in the MTS 858, the stress values cannot be accurately calculated; however, as in the quasi-static tests, the peak and valley strain were expected to be strongly correlated with the peak and valley capacitance. The first StretchFABRIC sensor exhibited moderate correlation between the peak and valley strain and capacitance returning a coefficient of 0.75, while the second sensor tested exhibited a poor correlation coefficient of 0.00124 (Table 4). Plots comparing the peak and valley strain and capacitance for the two sensors are provided in Figure 7A,B. Additionally, the peak displacement and the peak capacitance for the second sensor are separated (Figure 7B).

### 3.3. Participant Trials

Of the twelve participants recruited, ten participants completed the trials (10 Females, 1 Male). The fifth participant did not complete the trials because the cable connecting the sensor to the data puck was severed by catching on the top of a 3D printed spool. The first four trials were completed using the first fatigued StretchSense™ StretchFABRIC sensor, while the remaining six trials were completed using the second fatigued sensor.

#### 3.3.1. Dorsiflexion

Based on the results of the Shapiro–Wilk test, capacitance data for both the fatigued and unfatigued sensor were not normally distributed with both datasets returning a *p* = 0 (0 < 0.05 α); therefore, a two-tailed Mann–Whitney test (Wilcoxon Rank Sum test) (α = 0.05) was used to test the null hypothesis that the distribution of the fatigued sensors was equal to that of the unfatigued sensor [28]. Results of the Mann–Whitney test suggest that the distribution of capacitance between the fatigued and unfatigued capacitance datasets is statistically significantly different (asymptotic *p* = 0 < 0.05; reject the null hypothesis; distribution of the capacitance differs between the two datasets). The mean capacitance for the fatigued sensor (=430.9) trends 3.9% higher than the mean capacitance of the unfatigued sensor ($\bar{x}$ = 414.6) (Figure 8A,B).

#### 3.3.2. Plantarflexion

The mean capacitance of the fatigued sensor ($\bar{x}$ = 470.5) trended 2.7% higher than that of the unfatigued sensor in plantarflexion ($\bar{x}$ = 458.1) (Figure 9A,B). Based on the results of the Shapiro–Wilk test, the capacitance for neither sensor has a normal distribution with both datasets returning a *p* = 0 (0 < 0.05 α). Further, the results of a two-tailed Mann–Whitney test (α = 0.05) also suggest that the distributions of both the fatigued and unfatigued datasets are statistically significantly different (asymptotic *p* = 0 < 0.05).

#### 3.3.3. Eversion

The mean capacitance from the fatigued sensor ($\bar{x}$ = 422.1) trends 2.3% higher than that of the unfatigued sensor ($\bar{x}$ = 412.4) (Figure 10A,B). Based on the results of the Shapiro–Wilk test, neither the mean capacitance of the fatigued sensor, nor the mean capacitance of the unfatigued sensor are normally distributed with both returning *p* = 0 (0 < 0.05 α). A two-tailed Mann–Whitney test (α = 0.05) was again used to test the null hypothesis that the median capacitance from the fatigued sensor has the same distribution as that of the unfatigued sensor [28]. Results of the Mann–Whitney test suggest that the distributions of the two capacitance datasets are statistically significantly different (asymptotic *p* = 0 < 0.05; reject the null; the distributions of the two datasets differ).

#### 3.3.4. Inversion

Similar to dorsiflexion and eversion, the mean capacitance from the fatigued sensor ($\bar{x}$ = 447.7) trends 5.8% higher than the mean capacitance of the unfatigued sensor ($\bar{x}$ = 422.4) (Figure 11A,B). Results of the Shapiro–Wilk test reveal that the capacitance data for both sensors are not normally distributed. Both the fatigued and unfatigued capacitance datasets returned a *p* = 0 (0 < 0.05 α), and the results of the two-tailed Mann–Whitney test (α = 0.05) also suggest that the distributions of the two datasets are statistically significantly different (asymptotic *p* = 0 < 0.05; reject the null; the distributions of the two datasets differ).

#### 3.3.5. Gait Trials

Results of the gait trials were the same as those for the static movements. The mean capacitance of the fatigued sensor ($\bar{x}$ = 429.6) trended 1.7% higher than that of the unfatigued sensor ($\bar{x}$ = 422.3) (Figure 12A,B). Based on the results of the Shapiro–Wilk test (α = 0.05), neither capacitance dataset was normally distributed. The fatigued capacitance dataset returned a *p* = 0 < 0.05, while the unfatigued capacitance dataset returned a *p* = 3.77476 × 10^−15^. Based on the results of the two-tailed Mann–Whitney test (α = 0.05), the distributions of the datasets are statistically significantly different (asymptotic *p* = 0).

## 4. Discussion

Most fatigue studies of stretch sensors are focused on the LCF behavior of the sensors [14], with little explanation of the fatigue behavior of the signal [31]. For strain sensors to be useful during athletic practices and competitions, their HCF behavior must be understood [14]. After initial verification of the linear correlation between the strain and capacitance through quasi-static testing, two StretchSense sensors were subjected to HCF, applied to monitor the ankle motions that occur during activities of daily living, and their capacitive output was then compared to that of an unfatigued sensor when completing the same movements. Across all static and dynamic movements, the capacitance from the sensors subjected to 25,000 cycles was consistently higher than the capacitance from the unfatigued sensors, suggesting that the StretchSense™ StretchFABRIC sensors are subject to an upward drift over time. Whether the drift continues over time, or stabilizes at the higher capacitance upon completion of cyclic softening of the material, is unknown. Recent work based on a modification of Basquin’s equation for HCF [32], however, suggests that following a break-in period (i.e., cyclic softening) for the sensor, the signal will stabilize once the endurance limit is reached. The break-in periods ranged into hundreds to millions of cycles depending on the sensor tested [31]. Boland [31] argues that the modified equation can be used to predict when the endurance limit will be reached; however, Basquin’s equation [32] is dependent upon fully reversed loading conditions [32,33], and whether the equation accurately predicts the fatigue behavior of elastomers needs further investigation [34]. Further, the requirement of a break-in period for the signal implies that the data collected from a new “out-of-the-box” sensor may not be reliable raising the question of whether commercially sold stretch sensors should be precycled before reaching the end user.

Despite the upward drift, all of the capacitance values recorded from the fatigued sensor for the static movements fall within the 410–800 pF range of capacitance values previously recorded by the plantarflexion sensor during participant trials (Figure 4) [8]. For example, the capacitance for dorsiflexion peaked at 470 pF, while the capacitance for eversion peaked at approximately 437.5 pF (Figure 8 and Figure 10). Similarly, the peak capacitance for both plantarflexion and inversion also fell within the range of recorded values for the plantarflexion sensor, peaking at 575 pF and 518 pF, respectively (Figure 9 and Figure 11), initially suggesting that the effect of the drift at 25,000 cycles remains negligible; however, due to the small sample size, more research is needed to determine if the increase in capacitance is constant across all StretchFABRIC sensors and all participants, and to determine how well the fatigued capacitance correlates to joint angle. Further, whether the drift remains negligible as the cycling continues remains unknown, and if the sensors are to be used during the course of a sports season, an acceptable amount of drift in the capacitance should be determined. Once this amount of drift is exceeded, the sensor should be replaced. Additionally, of note, the data from strain sensors is considered reliable only when the sensor is in tension, but the results of the fatigue tests suggests that relaxation capacitance of the StretchFABRIC sensor may also be reliable, provided some modicum of tension is always held on the sensor (i.e., the sensor is never fully relaxed). Future research opportunities include subjecting the stretch sensors to more fatigue cycles to determine if the drift stabilizes and the development of a model to predict the amount of drift a sensor will incur during a particular use. Such a model incorporating both material and signal properties could be potentially developed using finite element methods and would be beneficial in determining when a sensor should be replaced.

## 5. Conclusions

Under quasi-static testing conditions, the strain of the material and the capacitive output of the stretch sensor are linearly correlated.Under HCF testing conditions, the strain of the material and the capacitive output are poorly to moderately correlated.When comparing the fatigued sensor to an unfatigued sensor during static and dynamic movements, the capacitance of the fatigued sensor consistently exhibits an upwards drift.More testing is needed to determine whether the upwards drift will stabilize once the polymers comprising the sensors have stabilized following cyclic softening.The results of these experiments suggest that the data collected from sensors “out of the box” may not be reliable, and that prestretching of the sensors by the manufacturer or end user may be required.

## Figures and Tables

**Figure 1 sensors-22-08168-f001:**
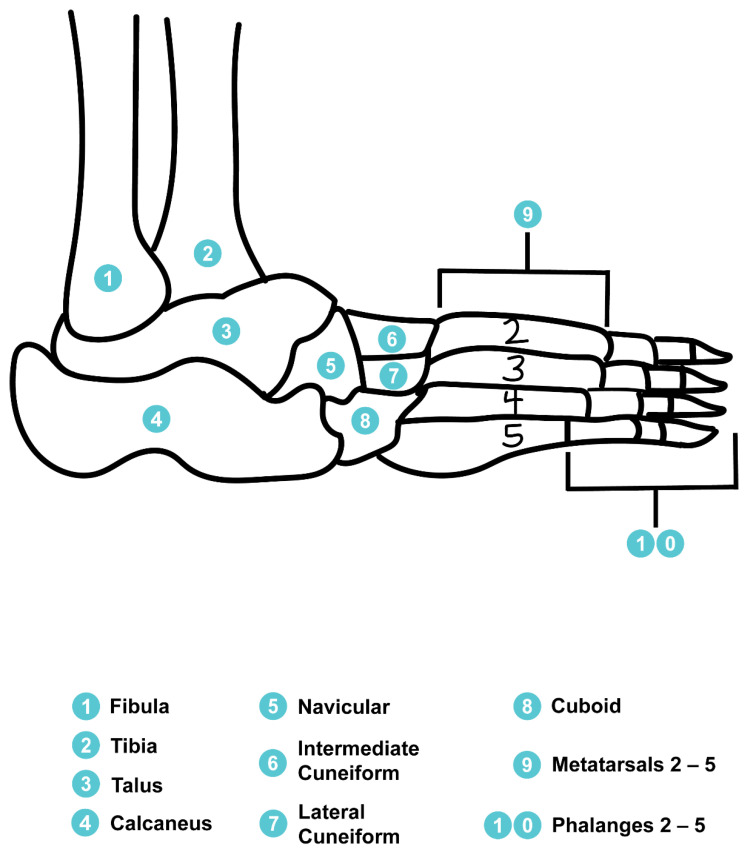
Lateral view of the osteology of the foot-ankle complex. Several bones including the medial cuneiform, first metatarsal, and first phalanges are not shown.

**Figure 2 sensors-22-08168-f002:**
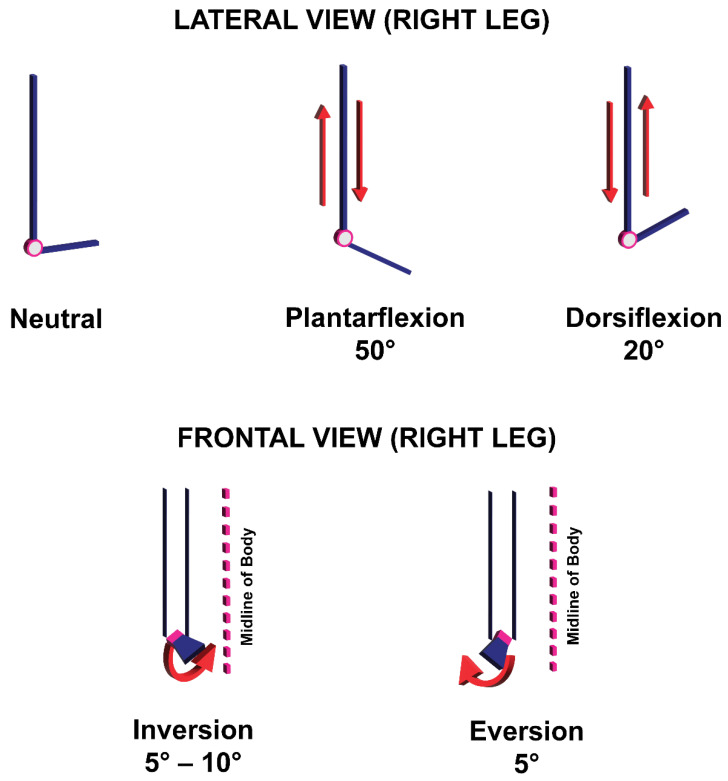
Maximum angles for movements produced by the human ankle joint.

**Figure 3 sensors-22-08168-f003:**
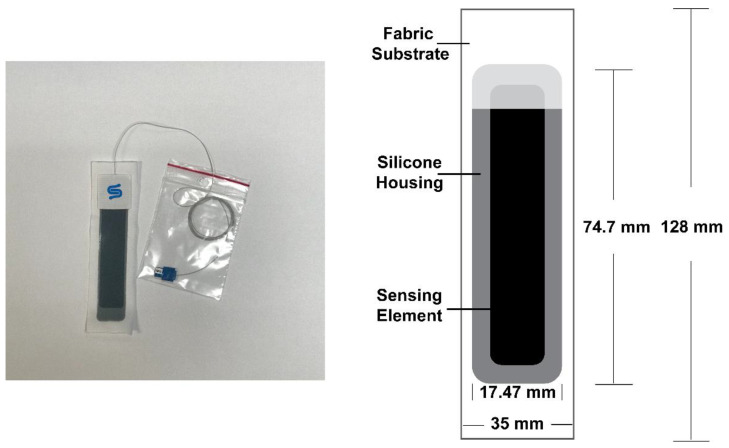
The StretchSENSE™ StretchFABRIC sensor is comprised of a proprietary sensing material that is encapsulated in silicone and attached to a fabric substrate. The sensor inclusive of the fabric substrate has a length of 128 mm, a width of 35 mm, and a thickness of 0.56 mm. Exclusive of the fabric substrate, the sensing element and its silicone housing have a length of 74.7 mm and a width of 17.47 mm. The fabric substrate is represented by the white rectangle, the silicone housing by the grey rectangle, and the sensing element is represented by the black rectangle.

**Figure 4 sensors-22-08168-f004:**
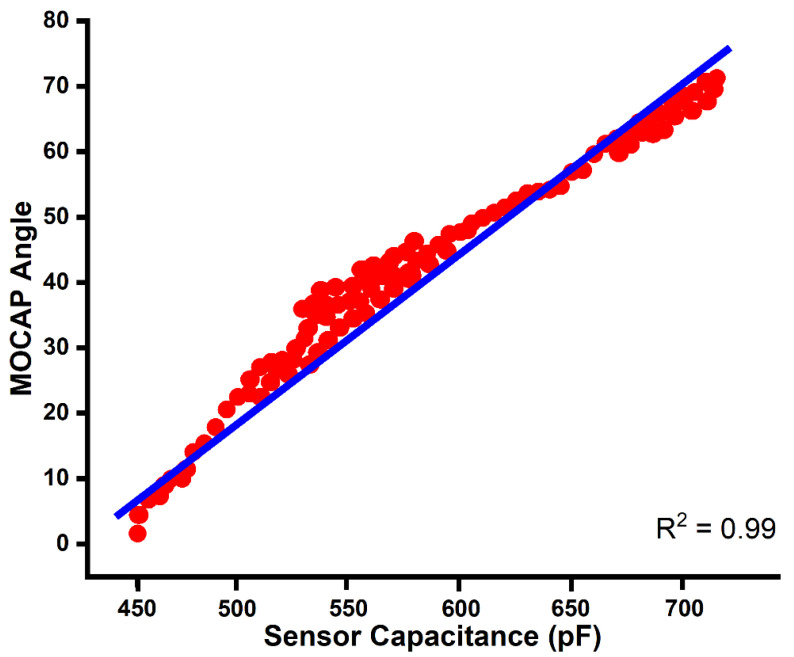
Comparison of joint angle to capacitance during plantarflexion of the foot-ankle complex. Motion capture (MOCAP) was used to determine the joint angle, while the capacitance data were collected using StretchSense™ StretchFABRIC sensors to produce this linear model correlating the two datasets. Adapted with permission of Saucier et al. [8], 2019, David Saucier.

**Figure 5 sensors-22-08168-f005:**
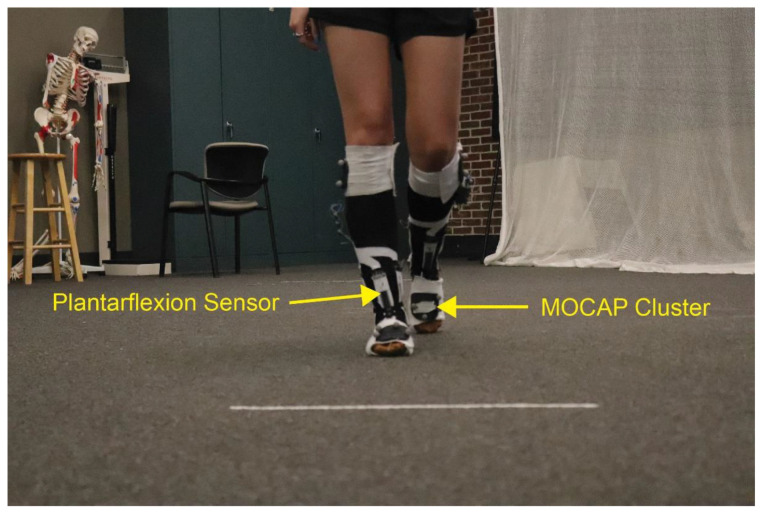
Example of the smart sock prototype in use during a gait trial.

**Figure 6 sensors-22-08168-f006:**
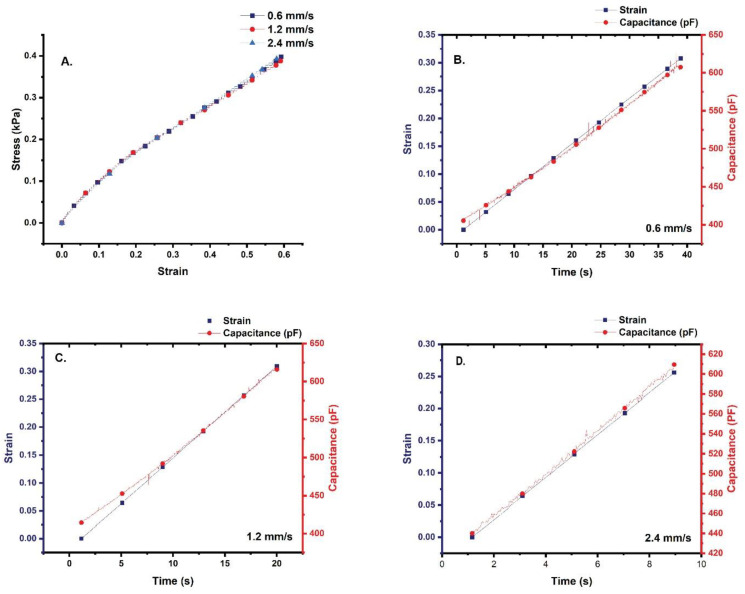
(**A**) Stress–strain curves produced from the quasi-static tensile testing of the stretch sensors. (**B**) Comparison of strain and capacitance at 0.6 mm/s. (**C**) Comparison of strain and capacitance at 1.2 mm/s. (**D**) Comparison of strain and capacitance at 2.4 mm/s. Note the differences in scale.

**Figure 7 sensors-22-08168-f007:**
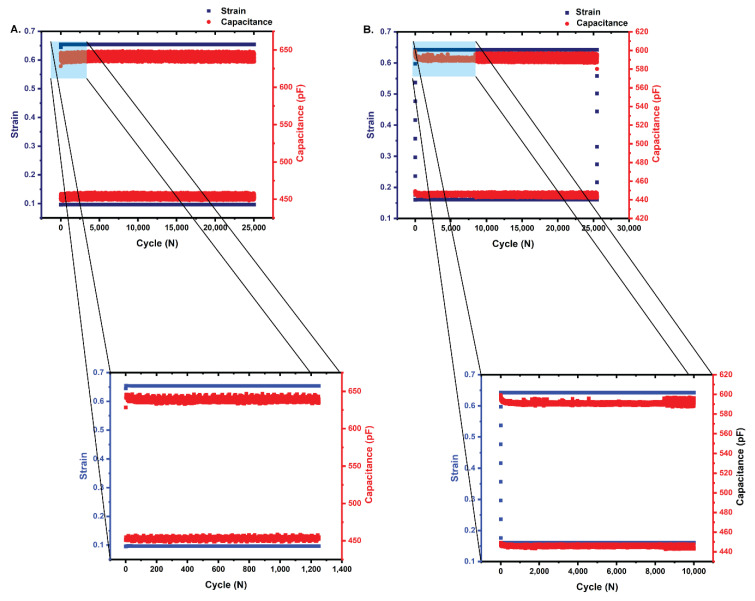
(**A**) Comparison of the peak and valley strain of the first fatigued StretchFABRIC sensor. The strain as measured by the MTS 858 and the capacitance as measured by the sensor are moderately correlated (Spearman’s rank correlation coefficient = 0.75) due to the lack of overlap between the strain and capacitance as shown in the close-up (blue box). (**B**) Comparison of the peak and valley strain of the second fatigued StretchFABRIC sensor. The strain as measured by the MTS 858 and the capacitance as measured by the sensor are poorly correlated (Spearman’s rank correlation coefficient = 0.00124) due to the variation in capacitance over the first 10,000 cycles as shown in the close-up (blue box). Note the separation of the peak strain and the peak capacitance. Additionally, note the different scales.

**Figure 8 sensors-22-08168-f008:**
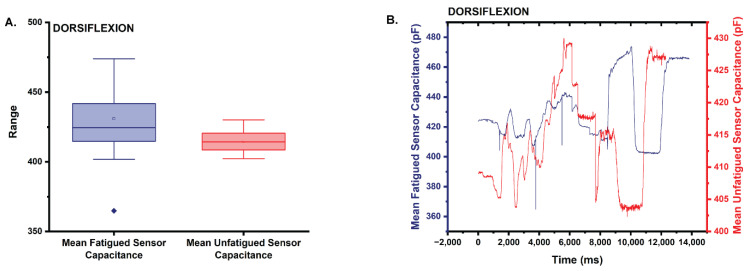
(**A**) Box plot of the statistics comparing the mean capacitance of the fatigued sensor to that of the unfatigued sensor in dorsiflexion. Note that the mean capacitance is higher for the fatigued sensor. (**B**) Plot comparing the mean capacitance of the fatigued sensor to the unfatigued sensor during the dorsiflexion trials.

**Figure 9 sensors-22-08168-f009:**
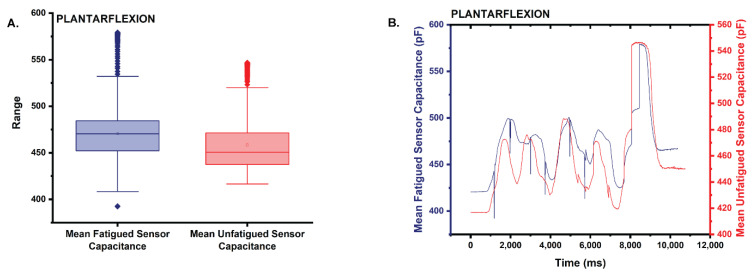
(**A**) Box plot of the statistics comparing the mean capacitance of the fatigued sensor to that of the unfatigued sensor in plantarflexion. Note that the mean capacitance is higher for the fatigued sensor. (**B**) Plot comparing the mean capacitance of the fatigued sensor to the unfatigued sensor during the plantarflexion trials.

**Figure 10 sensors-22-08168-f010:**
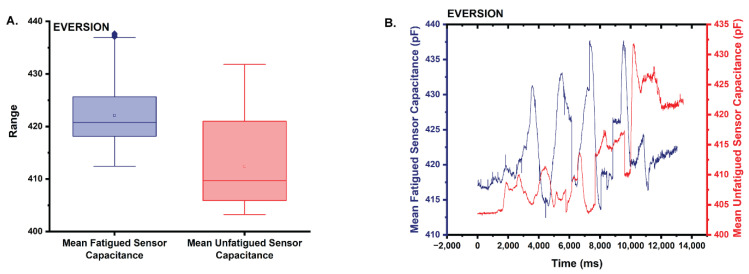
(**A**) Box plot of the statistics comparing the mean capacitance of the fatigued sensor to that of the unfatigued sensor in eversion. Note that the mean capacitance is higher for the fatigued sensor. (**B**) Plot comparing the mean capacitance of the fatigued sensor to the unfatigued sensor during the eversion trials.

**Figure 11 sensors-22-08168-f011:**
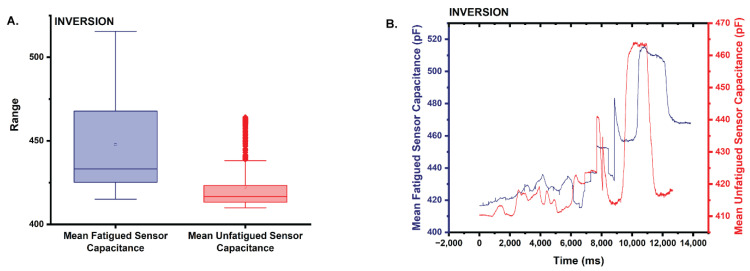
(**A**) Box plot of the statistics comparing the mean capacitance of the fatigued sensor to that of the unfatigued sensor in inversion. Note that the mean capacitance is higher for the fatigued sensor. (**B**) Plot comparing the mean capacitance of the fatigued sensor to the unfatigued sensor during the inversion trials.

**Figure 12 sensors-22-08168-f012:**
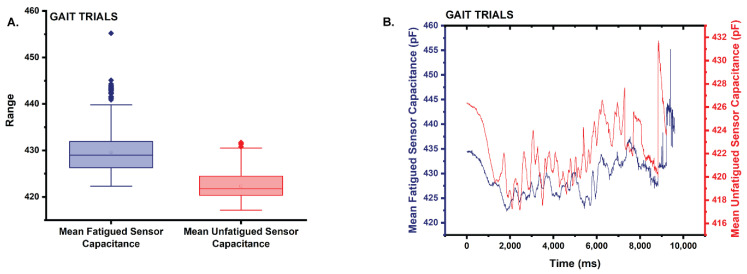
(**A**) Box plot of the statistics comparing the mean capacitance of the fatigued sensor to that of the unfatigued sensor during the gait trials. Note that the mean capacitance is higher for the fatigued sensor. (**B**) Plot comparing the mean capacitance of the fatigued sensor to the unfatigued sensor during the gait trials.

**Table 1 sensors-22-08168-t001:** Results of the Shapiro–Wilk Test of Normality for the Quasi-Static Data.

Speed	Data	*p*-Value	Decision
0.6 mm/s			
	Capacitance	1.11022 × 10^−16^	*p* < 0.05; reject null; data are not normally distributed
	Strain	0	*p* < 0.05; reject null; data are not normally distributed
1.2 mm/s			
	Capacitance	1.68389 × 10^−11^	*p* < 0.05; reject null; data are not normally distributed
	Strain	3.33067 × 10^−16^	*p* < 0.05; reject null; data are not normally distributed
2.4 mm/s			
	Capacitance	3.00829 × 10^−6^	*p* < 0.05; reject null; data are not normally distributed
	Strain	1.42418 × 10^−10^	*p* < 0.05; reject null; data are not normally distributed

**Table 2 sensors-22-08168-t002:** Results of the Lilliefors Test of Normality for the Fatigue Test Data.

Sensor	Data	*p*-Value	Decision
1			
	Capacitance	0	*p* < 0.05; reject null; data are not normally distributed
	Strain	0	*p* < 0.05; reject null; data are not normally distributed
2			
	Capacitance	0	*p* < 0.05; reject null; data are not normally distributed
	Strain	0	*p* < 0.05; reject null; data are not normally distributed

**Table 3 sensors-22-08168-t003:** Spearman’s Rank Correlation Coefficients for the Quasi-Static Strain and Capacitance at Each Speed.

Speed		Strain	Capacitance
0.6 mm/s		Spearman’s Correlation	Spearman’s Correlation
	Strain	1	0.99973
	Capacitance	0.99973	1
1.2 mm/s			
	Strain	1	0.99982
	Capacitance	0.99982	1
2.4 mm/s			
	Strain	1	0.9994
	Capacitance	0.9994	1

The strain was measured by the MTS 858, while the capacitance was measured by the StretchSense sensor, and both values are expected to be strongly correlated (Spearman’s Rank near or equal to 1).

**Table 4 sensors-22-08168-t004:** Spearman’s Rank Correlation Coefficients for the Peak and Valley Strain and Capacitance for the Two Fatigued StretchFABRIC sensors.

Sensor		Strain	Capacitance
1		Spearman’s Correlation	Spearman’s Correlation
	Strain	1	0.75137
	Capacitance	0.75137	0.99998
2			
	Strain	1	0.0169
	Capacitance	0.0169	1

The strain was measured by the MTS 858, while the capacitance was measured by the StretchSense sensor. The peak and valley strain and the peak and valley capacitance were expected to be strongly correlated (Spearman’s Rank near or equal to 1).

## Data Availability

Not applicable.

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
