# Peer review of "Comparison of the Capacitance of a Cyclically Fatigued Stretch Sensor to a Non-Fatigued Stretch Sensor When Performing Static and Dynamic Foot-Ankle Motions"

_sensors, 2022, doi:10.3390/s22218168_

Round 1

Reviewer 1 Report

The present paper covers a very interesting field (capacitive sensor for biomedical applications) and the findings are really interesting too , showing the difference in performance between a pre fatigued sensor and a non pre fatigued one. In the reviewer's opinion the paper should be accepted with minor revision: if the following comments are addressed the paper can surely be published.

1) 3D printing is such a good technology to fabricate soft structures with sensors, in recent years several attempts to fabricate capacitive sensors have been done. You guys should mention that. three great papers showing that are:

1) Ragolia, Mattia Alessandro, et al. "Thermal characterization of new 3d-printed bendable, coplanar capacitive sensors." Sensors 21.19 (2021): 6324.; 2)Shemelya, Corey, et al. "Encapsulated copper wire and copper mesh capacitive sensing for 3-D printing applications." IEEE Sensors Journal 15.2 (2014): 1280-1286; 3)Santiago, Carolyn Carradero, et al. "3D printed elastomeric lattices with embedded deformation sensing." IEEE Access 8 (2020): 41394-41402, and 4) Loh, Leon Yeong Wei, et al. "3D printed metamaterial capacitive sensing array for universal jamming gripper and human joint wearables." Advanced Engineering Materials 23.5 (2021): 2001082.   2) I would suggest adding a figure of the actual sensor in fig 3   3) The conclusion section is missing. I know there is a good discussion section but I believe the authors could benefit from a short conclusion section where the main findings are listed as bullet points    

Author Response

Thank you for your review. We have updated the manuscript to include the papers that you suggested in lines 54-56, and we have also added a conclusions section beginning on line 486.

Reviewer 2 Report

Thank you for submitting this manuscript.  I found this product presented very well. 

The following comments are merely observations, as I did not find anything of substance in need of refinement worth mentioning in the body of the manuscript. 

- the authorship ends with an initial. This is inconsistent with the suggested format and the format of the other authors listed prior. 

- the discussion of the manuscript ends abruptly. There is no really closing point, merely a statement. As this section is a bit stout in nature (recognizing the intent of the manuscript), perhaps a mention of future application or direction of the research would allow the consumer of the information to advance their thoughts on the topic. 

Overall, I found the manuscript constructed solidly and presented in a manner easy to consume as well as very interesting. 

Author Response

Thank you for your review. We have added options for future research at the close of our discussion beginning with line 479.

Reviewer 3 Report

In the paper, a smart sock that is equipped with stretch sensors to record movement data outside of the laboratory. To test the durability of the stretch sensors, the sensors were exposed to high-cycle fatigue testing with simultaneous capture of the capacitance. Following randomization, either the fatigued sensor or an unfatigued sensor was placed in the plantarflexion position on the smart sock, and participants were asked to complete the following static movements: dorsiflexion, inversion, eversion, and plantarflexion. Results of the tests show that the capacitive output of the fatigued sensor was consistently higher than that of the unfatigued sensor for both the static and dynamic movements, suggesting that an upwards drift of the capacitance was occurring in the fatigued sensors.

Over all, this manuscript is well organized in terms of design, testing and data analysis, and the concluding remark is consistent to the results. I think this paper is quite interesting in the community of basketball sports training and health monitoring by application of wearable sensors, for aiding the doctors diagnosis and footwear development. It should be acceptable for publishing in this special issue addressing some minor revision as following.

1.       In the section of Introduction, the title to the first sub-section 1.1 was not found.

2.       Since the authors claim the application of the commercially available sensors (StretchSENSE™ StretchFABRIC sensor), please provide more details about the function and working principle of the sensors utilized for the study, in order to explicitly show the novelty of the investigation.

3.       As is shown in Figure 7, it is difficult for the readers to figure out the correlation of two different sensors. For instance, how can you identify the distinct values of the Spearman’s rank correlation from this figure? Please give some more explanation about Spearman’s Correlation shown in the Tables as well for better demonstration.

4.       Line 88 and Line 91, the numbers 1260 steps and 40320 steps is quite unbelievable for a normal basketball match. Additionally, the data in Figure 4 is too straight, is that true? Please confirm the above points.

5.       The reason for doing the quasi-static test using the fatigued sensor and unfatigued sensor is not very clear. Besides, it is said in Line 420, “The two sensors used in this study were exposed to HCF and then applied to monitor the ankle motions that occur during activities of daily living”. It is confusing for me a little bit here that were both sensors exposed to high-cycle fatigue (HCF)? If so, what does that mean by “unfatigued”? Please address this point.

Author Response

Thank you for your review. The subsections have been renumbered, and we have included more information regarding the StretchSense sensors in lines 164-168. In figure 7, we have included a zoomed in view to the first 1000-10000 cycles to show the difference between the strain and capacitance values. Footnotes have also been added to the tables to clarify the origins of the values. The number of steps that a player could incur over a season is based on the literature where the number of steps were counted. Figure 4 has been redigitized to include more of the data points to show the linear model fit. The purpose of the quasi-static testing has been added in lines 148-149. Lines 443-447 have also been added to clarify the the difference between the quasi-static and fatigue tests.